# Response of Plant Endophyte Communities to Heavy Metal Stress and Plant Growth Promotion by the Endophyte *Serratia marcescens* (Strain JG1)

**DOI:** 10.3390/plants13192755

**Published:** 2024-09-30

**Authors:** Jiayi Liu, Chao Liu, Jiaxin Zheng, Xiaoxia Zhang, Kang Zheng, Jiayao Zhuang

**Affiliations:** 1Collaborative Innovation Center of Sustainable Forestry in Southern China of Jiangsu Province, Nanjing Forestry University, Nanjing 210037, China; 2China Construction First Group Co., Ltd., Beijing 100000, China

**Keywords:** tailings, heavy metal, endophytic communities, growth-promoting effect

## Abstract

Effects of heavy metals on soil microbial communities have been extensively studied due to their persistence in the environment and imposed threats to living organisms; however, there is a lack of in-depth studies of the impacts of heavy metals on plant endophyte communities. Therefore, the responses of plant endophyte communities to different concentrations of heavy metals were investigated in this study. The endophyte communities of plants existing in severely (W1, Pb, 110.49 mg/kg, Cd, 1.11 mg/kg), moderately (W2, Pb, 55.06 mg/kg, Cd, 0.48 mg/kg), and mildly (W3, Pb, 39.06 mg/kg, Cd, 0.20 mg/kg) contaminated soils were analyzed by 16s rRNA high-throughput Illumina sequencing. Furthermore, networks were constructed to illustrate the relationships between microorganisms and environmental factors. High-quality sequences were clustered at a 97% similarity level. Results revealed that the diversity of the community and relative abundance of Cyanobacteria phylum increased with decreasing levels of pollution. Cyanobacteria and Proteobacteria were found to be the dominant phylum, while *Methylobacterium* and *Sphingomonas* were observed as the dominant genus. Tukey’s HSD test showed that the relative abundances of Cyanobacteria and Proteobacteria phyla and *Methylobacterium* and *Sphingomonas* genera differed significantly (*p* < 0.01) among the plants of the three sample sites. Environmental factor analysis revealed a significant negative correlation (*p* < 0.01) of Cyanobacteria and a significant positive correlation (*p* < 0.01) of *Methylobacterium* with the heavy metal content in the environment. These findings suggest that Cyanobacteria and *Methylobacterium* may be phylum and genus indicators, respectively, of heavy metal toxicity. Tax4Fun analysis showed the effect of heavy metal toxicity on the abundance of genes involved in plant metabolism. In addition, culturable endophytic strains were isolated to study their resistance to heavy metal stress and their ability to promote plant growth. The potting tests showed that the JG1 strain was tolerant to heavy metals, and it could significantly promote the growth of the host plant under stress caused by multiple heavy metals. Compared to the control, the JG1-treated plants showed a 23.14% increase in height and a 12.84% increase in biomass. Moreover, AP, AK, and HN contents in JG1-treated plants were 20.87%, 12.55%, and 9.03% higher, respectively, under heavy metal stress. The results of this study provide a scientific basis for the construction of an efficient plant endophyte restoration system.

## 1. Introduction

Chemicals, smelting, mining, and other human activities produce a large number of heavy metals, which are discharged into the soil environment directly or indirectly, causing serious pollution and posing threats to plants and human health [1,2]. Heavy metals (HMs), such as Pb and Cd, are highly toxic and have adverse effects on plants, microorganisms, and ecosystems [3]. At present, soil remediation research is mainly focused on phytoremediation and microbial remediation. Among these methods, phytoremediation has attracted much attention. It is an effective method to eliminate the heavy metals in soils and restore the ecological balance in the mined areas. Furthermore, this process is cost-effective, environment-friendly, and sustainable [4]. However, the practical application of this technique is difficult due to the long treatment cycle. Moreover, the efficiency of phytoremediation is easily affected by the morphology of heavy metals, soil conditions, and selective absorption by plants [5,6]. On the other hand, microbial remediation offers the advantages of low cost and high efficiency, and it has great potential in soil remediation. However, microbial remediation research is still limited to the laboratory scale since microorganisms are small and difficult to separate from the environment [7,8]. Therefore, researchers are now exploring combined endophyte–microbial remediation to mitigate heavy metal contamination in soil.

Bacterial endophytes commonly exist in healthy plant tissues without causing any harm to the host plant [9]. Bacterial endophytes can increase their biomass by regulating plant growth regulators. Moreover, bacterial endophytes can dilute heavy metals in certain parts to reduce their toxic effects on plants. Badu found that *Bacillus thuringiensis* GDB-1 promoted the accumulation of heavy metals in plants under heavy metal stress by promoting the synthesis of ACC deaminase and increasing the plant biomass [10]. Furthermore, biosurfactants produced by bacterial endophytes can also improve the bioavailability of insoluble heavy metals, thereby accelerating the rate of phytoremediation [11].

*Artemisia argyi H. Lév. & Vaniot.* is a predominant plant species in lead–zinc mining areas [12], which is highly tolerant to heavy metals (Pb, Cd, and Mn) and shows rapid growth and increase in biomass [13,14]. Therefore, it may be an ideal plant species for phytoremediation. As an extremely resilient shrub, *Amorpha fruticosa* L. plays an important role in the phytoremediation of heavy metal contaminated areas due to its low cost and excellent slope protection properties. In previous studies, *Amorpha fruticosa* L. has shown strong tolerance to heavy metals and the ability of heavy metal enrichment [15,16]. This species can grow in heavy metal-polluted soil, alleviating the heavy metal stress in soil environments [17,18]. However, only a few studies have explored the resistance of *Amorpha* L. to heavy metals such as Pb and Cd. To understand the response strategies of endophytic communities to varying concentrations of Pb and Cd pollution, several *Artemisia argyi H. Lév. & Vaniot.* plants were collected from the contaminated areas with different concentrations of heavy metals. The endophytic communities of the collected plants were analyzed by Illumina high-throughput sequencing. Simultaneously, culturable endophytic bacteria were isolated, and their effects on the growth of *Amorpha fruticosa* L. were studied. Furthermore, the physical and chemical properties of contaminated soils were analyzed to determine the relationships between endophytic communities and environmental factors. The findings of this study can be used as a theoretical reference for phytoremediation in heavy metal contaminated areas.

## 2. Results

### 2.1. Plant and Soil Properties

In this study, the evaluation standard adopts the screening value and control value of soil pollution risk of agricultural land in the Soil Environmental Quality Agricultural Soil Pollution Risk Control Standard (Trial) [19] as the evaluation standard of heavy metal pollution in agricultural land, and at the same time, combined with the background value of the soil elements in Shanxi Province, to judge and further refine the evaluation process; and finally, based on the size of the Nemerow Integrated Pollution Index to reflect the degree of heavy metal contamination of the surface soil in the study area. Table 1 shows the concentrations of Pb and Cd in the rhizosphere soil of the three plots and the corresponding plant growth status. The table indicates varying degrees of heavy metal pollution in the rhizosphere soil of the three plots, with the highest content of Pb and Cd found in the W1 plot and the lowest in the W3 plot. In addition, the concentrations of Pb and Cd exceeded the background values of Shanxi Province, but the content of Pb was lower than that of the soil environmental quality of China [19], and the average concentration of Cd was close to twice the risk screening value. Nemerow comprehensive pollution index (P_N_) shows that W1 P_N_ 5.76 > 3 is a severely polluted area, W2 P_N_ 2.81 is a moderately polluted area, and W3 P_N_ 1.89 is a mild polluted area.

Furthermore, as heavy metal content in soil decreases, plant height and biomass increase to varying degrees. This suggests that heavy metal pollution can indirectly impact plant growth by altering the physical and chemical properties of soil.

Figure 1A displays the average heavy metal contents in various parts of *Artemisia argyi H. Lév. & Vaniot.* plants collected from different contaminated sites. The comparison between the heavy metal content in plants and the corresponding soil environments revealed that the levels of Pb and Cd in the plants collected from the three sites were consistent with the Pb and Cd contents in the corresponding rhizosphere soil samples. Furthermore, the concentration of Pb was higher than the Cd content in plants. Additionally, the concentration of Pb was higher in the roots compared to the shoots, while Cd content in the roots was lower than that in the shoots.

BCF indicates the ability of plants to accumulate elements from the soil (Figure 1B). TF roughly reflects the ability of heavy metals to translocate from the roots to the aerial parts of plants (Figure 1C). The figures show that *Artemisia argyi H. Lév. & Vaniot.* plants had lower Pb enrichment and transport abilities compared to Cd. The BCF of plants to Pb initially increased and then decreased with the decreasing concentration of heavy metals, while TF first decreased and then increased. On the other hand, BCF and TF of Cd in plants increased continuously with the decreasing level of heavy metals. BCF and TF of Cd were greater than 1. The results indicate that *Artemisia argyi H. Lév. & Vaniot.* has a high potential to enrich and transport Cd, even under varying concentrations of heavy metals in soil. However, its capacity to enrich and transport Pb is relatively weak.

### 2.2. Effects of HM on Plant Bacterial Communities

The bacterial communities of nine plant samples were analyzed using Illumina sequencing. The dataset of nine samples consisted of 609,594 unique 16S rDNA gene tags. Bacterial abundance in plant samples was assessed through α diversity (ACE, Chao1, Shannon, and Simpson) analysis (Figure 2). The ACE and Chao1 indices were used to determine the predicted number of OTUs in the plant samples, which reflects the species richness of the sample communities. The Shannon and Simpson indices were used to evaluate the evenness and richness of microbial communities. The figure illustrates that α diversity increased as heavy metal content decreased, with the lowest value observed in sample W1.

Based on the Bray–Curtis distance between the samples, NMDS and PCoA analyses of the plant bacterial communities in the three sample sites were performed using the R language (Figure 3). NMDS ordination shows the existence of significant differences in the composition of entire bacterial communities at the OTU level between the three types of areas (stress = 0.000, *p* < 0.05), especially between the tailing area and undisturbed area and between the remediation area and undisturbed area (Figure 3A), which is also clearly indicated by the 3D PCoA pattern (PCo1, PCo2, and PCo3 explained 80.87%, 9.49%, and 5.84% of the variance, respectively) (Figure 3B). These results indicate that there was a significant difference in the composition of plant endophyte communities between the three types of habitats.

Figure 4 displays the bacterial community structure in plant samples at the phylum and genus levels. The dominant phyla in the plant samples were Cyanobacteria and Proteobacteria (Figure 4A), accounting for over 60% of the total abundance. The relative abundance of Cyanobacteria (25.44%) was significantly lower than that of W2 (65.59%) and W3 (80.68%) in sample plot W1, while the relative abundance of Proteobacteria (42.92%) was significantly higher than that of W2 (30.89%) and W3 (12.85%). *Methylobacterium*, *Sphingomonas*, and *Hymenobacter* were the dominant genus in the plant samples (Figure 4B), with their relative abundance being significantly higher in W1 (13.96%, 14.32%, and 14.23%, respectively) compared to W2 (8.46%, 3.93%, and 0.56%) and W3 (1.27%, 0.87%, and 0.77%).

Tukey’s HSD test was utilized to determine the significance of species differences at both the phylum (Figure 4C) and genus levels (Figure 4D). The relative abundance of Cyanobacteria and Proteobacteria varied significantly (*p* < 0.01) among the three sample plants, while *Methylobacterium* and *Sphingomonas* also exhibited significant differences (*p* < 0.01). It indicated that Cyanobacteria and Proteobacteria and *Methylobacterium* and *Sphingomonas* may be related to the difference in heavy metal content in plants.

Pearson correlation analysis verified the correlation between environmental heavy metal content and plant physiological characteristics with plant endophyte communities (Figure 5). At the phylum level (Figure 5A), Cyanobacteria showed a significant negative correlation (*p* < 0.01) with environmental heavy metal content and a significant positive correlation (*p* < 0.01) with plant height and biomass. Proteobacteria showed a significant positive correlation (*p* < 0.01) with environmental heavy metal content and a significant negative correlation (*p* < 0.01). At the genus level (Figure 5B), *Methylobacterium* and *Sphingomonas* showed a significant positive correlation (*p* < 0.01) with environmental heavy metal content and a significant negative correlation (*p* < 0.01) with plant height and biomass. In contrast, Cyanobacteria and *Methylobacterium* showed a more significant correlation with heavy metal content in plants and soil.

The study determined the functional abundance of bacterial communities in plants from three sample plots using 16S rDNA gene amplicon data and Tax4Fun (Figure 6). Several pathways were significantly enriched in the phytobacterial community in sample plot W3 (*p* < 0.05), including genes associated with metabolism (carbohydrate metabolism, lipid metabolism, metabolism of cofactors and vitamins, and so on), genetic information processing (folding, sorting, degradation, replication, and repair), and organismal systems (endocrine system). Furthermore, the phytobacterial community in the W3 sample showed significant enrichment in nearly all secondary metabolic pathways. In addition, the abundance of bacterial metabolic functions varied among plants from different contaminated sites, depending on the concentration of heavy metals. The data indicate that there is a correlation between high microbial metabolism and low heavy metal concentrations in plants. Additionally, the study found that robust microbial metabolism was linked to decreased levels of heavy metals in both soil and plants.

### 2.3. Isolation and Identification of Plant Endophytes and Plant Growth Promotion Capability

A total of 30 culturable bacterial endophytes were isolated. Among them, 12 strains with growth-promoting effects were screened out through plate culture experiments of dissolving phosphorus, fixing nitrogen, and dissolving potassium. The results of the pot test showed that JG1 had the most significant plant growth promotion ability. JG1 was identified as *Serratia marcescens* FZSF02. Figure 7 shows colony growth images and cell morphology electron microscopy images.

Plant growth characteristics are demonstrated in Figure 8 and Table 2. The results show that the addition of JG1 significantly increased plant height by 54.19%, mean leaf area by 71.87%, root length and root volume by 33.87%, and plant biomass by 33.87% in the absence of heavy metals. This shows that JG1 has a significant growth-promoting effect under a heavy metal-free environment. The addition of heavy metals significantly optimized the growth of plants supplemented with JG1 compared to CK. Plant height increased by 23.14%, ground diameter increased significantly by 39.25%, and average leaf area increased significantly by 59.71%. Although the root-related indicators decreased, the plant biomass increased by 12.84%. This shows that JG1 still has a strong promoting effect on the growth of *Amorpha fruticose* L. plants in a heavy metal environment.

Soil nutrient concentrations are demonstrated in Figure 9. The figure shows that JG1 significantly improved the physical and chemical properties of the soil. In the treatment group where heavy metals were not added, the soil’s AP, AK, and HN were significantly increased by 36.70%, 20.41%, and 24.52%, respectively, compared to the CK. The pH did not change significantly. After the addition of heavy metals, the JG1 treatment group showed a general increase of 20.87% in AP, 12.55% in AK, and 9.03% in HN compared to the CK group. Additionally, the pH increased from 6.82 to 6.88. These results suggest that JG1 has a positive effect on soil physical and chemical properties.

## 3. Discussion

### 3.1. Effects of Different Pollution Levels of Tailings on the Growth of Local Dominant Plants and Heavy Metal Concentration

External environmental conditions usually constrain the growth of plants around the mining areas [20,21]. However, native dominant plant species in these areas have developed tolerance and ability to absorb and accumulate heavy metals. This has resulted in the generation of pollution-resistant ecotypes. Some species have even evolved to become heavy-metal super-enriched plants [22]. In this study, the dominance of *Artemisia argyi H. Lév. & Vaniot.* was observed in the contaminated areas, and the biomass and height of plants increased with the decreasing level of contamination. This finding is consistent with the results reported by Pan [23]. Furthermore, the plants in all three polluted sites exhibited a higher concentration of Pb than Cd, which may be attributed to the variations in the total heavy metal concentration and bioavailability in the soil, which were consistent with Liu’s findings [24].

BCF and TF are important indicators of the uptake and transport capacity of plants for Pb and Cd. In this study, *Artemisia argyi H. Lév. & Vaniot.* showed lower Pb enrichment in the shoots, as compared to the roots, and TF was less than 1. This was mainly due to the presence of Pb in ionic states (Pb(PO_4_)_2_ and PbCO_3_) and complex states [25]. Due to the suction and retention, as well as passivation and precipitation, it was difficult to transport the absorbed Pb to the aboveground part of the plant root system. Therefore, most of the absorbed Pb by the plant was confined in the roots, and the Pb enrichment capacity of *Artemisia argyi H. Lév. & Vaniot.* was poorer [26]. On the contrary, Cd uptake and translocation capacity of *Artemisia argyi H. Lév. & Vaniot.* was strong, as indicated by its BCF and TF values, which were greater than 1. Furthermore, the BCF and TF values increased with decreasing Cd concentration in soil, which was consistent with the findings reported by previous research [27].

### 3.2. Effects of Different Heavy Metal Concentrations on Endophytic Bacteria Community

Heavy metal concentrations in plants and soils are important factors that affect plant growth and the abundance and diversity of microbial communities. The α diversity of endophytic communities in nine plant samples were analyzed. The results revealed that the α diversity was significantly affected by the concentration of heavy metals in the environment. A decrease in species diversity and richness was observed with the increase in the concentration of heavy metals in the environment, which was similar to the microbial diversity trend reported by Wang [28]. This finding indicates that the increasing environmental stress affects the structure and distribution of microbial communities in the environment. Furthermore, PCoA analysis showed that habitat also had a significant impact on the composition of the endophyte community. Overall, heavy metal concentration and increased sample heterogeneity were found to be the main factors affecting the distribution of bacterial endophytes.

Heavy metal pollution typically reduces the microbial diversity in soil. However, if the heavy metals persist in soil for an extended period, the microorganisms develop tolerance to these pollutants, which results in a subsequent increase in the diversity and richness of certain specific microorganisms [29], even at low concentrations of heavy metals [30]. In this study, Proteobacteria was most abundant in heavily polluted sites, which is consistent with previous research reporting that the Proteobacteria phylum shows high tolerance to heavy metals, and it is the most common phylum in the soils contaminated by high concentrations of heavy metals [31]. Furthermore, Cyanobacteria abundance increased with the decrease in heavy metal concentration, and it ultimately became the dominant phylum. This finding is in line with the results reported by Lin [32]. Cyanobacteria are known for their tolerance to metal ions and their ability to adsorb heavy metals by producing extracellular substances, such as polysaccharides. Additionally, the oxygen produced by Cyanobacteria through photosynthesis can promote plant growth [33].

*Methylobacterium* has been reported to be tolerant to Cd, Pb, and AS. These bacteria can promote plant growth by immobilizing heavy metals [34]. In recent years, metal-resistant *Methylobacterium* species have also been discovered in plant endophyte communities. These resistant strains can not only enhance the resistance of plants to stress but also improve the ability of plants to accumulate heavy metals [35]. Therefore, the increased abundance of *Methylobacterium* species may be the reason for plant growth under high heavy metal stress in this study. Furthermore, *Sphingomonas* can promote plant growth by producing growth hormones and increasing the stress tolerance of plants. For instance, *Sphingomonas* LK11, isolated from plant leaves, produced auxin and gibberellins, which promoted the growth of Solanum lycopersicum L. and increased its tolerance to Cd stress [36]. Research has shown that bacterial endophytes can increase the accumulation of heavy metals in host plants. For example, the host plants inoculated with *Sphingomonas* SaMR12 showed higher accumulation of Cd and Zn [37]. In this study, *Methylobacterium* and *Sphingomonas* were observed as the dominant genera, which promoted the enrichment of heavy metals in plants and maintained plant growth under high heavy metal stress.

### 3.3. Plant Growth-Promoting Benefits of JG1 (Serratia marcescens)

JG1 exhibited a high tolerance to heavy metals. Inoculation with JG1 resulted in a significant increase in plant height, ground diameter, average leaf area, root length, root volume, and plant biomass, as compared to the control. Additionally, JG1 increased the nutrient levels in the soil. This increase remained significant even under heavy metal stress. Almaghrabi discovered that *Serratia marcescens*. isolated from plants or plant rhizospheres could promote the growth and development of plants by decomposing cell walls and releasing a large amount of indole acetic acid, phosphate, cellulose, and pectinase [38]. Shylla discovered that *Serratia marcescens* can adsorb up to 83% of Pb and effectively reduce the acidity of the medium [39]. Furthermore, Cocozza reported that *Serratia marcescens* promoted the growth of *Populus* L., which, in turn, improved the Cd tolerance of the plant [40]. These studies suggest that *Serratia marcescens* may play a crucial role in the adaptation of host plants to higher concentrations of heavy metals by promoting their growth and inducing systemic responses. However, further research is required to determine the exact mechanism behind the high plant growth under heavy metal stress.

## 4. Materials and Methods

### 4.1. Study Sites and Sampling

The sampling sites were located in the Pb mine in Lingqiu County, Shanxi Prov Linneo ince, China (114°12′41″ E, 39°21′30″ N). In October 2021, three samples of *Artemisia argyi H. Lév. & Vaniot.* plants were randomly collected from areas with severe (W1, Pb, 110.49 mg/kg, Cd, 1.11 mg/kg), moderate (W2, Pb, 55.06 mg/kg, Cd 0.48 mg/kg), and mild (W3, Pb, 39.06 mg/kg, Cd, 0.20 mg/kg) Pb–Cd pollution, and their plant height was measured (each plant at least 5–10 m away from other plants). At the same time, plant rhizosphere soil (about 5–10 cm deep) was collected. The samples from each plot were fully mixed as one sample, resulting in a total of 9 samples. Each sample was placed in a sterile plastic bag and stored in an incubator (with ice packs), then transported to the lab and processed within 24 h.

### 4.2. Plant and Soil Analysis

Soil samples were air dried (25 °C), then crushed and sieved through a 20 mesh to obtain a fine powder. For HM concentration analysis, soil powders (0.5 g) were digested with a 4 mL HCl/HNO_3_ (3:1, *v*/*v*) mixture at 80 °C for 30 min and 100 °C for 30 min, followed by 120 °C for 1 h, then cooled, and 1 mL HClO_4_ was added to continue digestion at 100 °C for 20 min, then 120 °C for 1 h. Finally, the digests were diluted to 50 mL with triple deionized water in a volumetric flask. The available Pb and Cd in the soils were extracted with diethylenetriamine pentaacetic acid triethanolamine (DTPA-TEA). The Nemerow Integrated Pollution Index (P_N_) method was used to determine the risk level of soil heavy metal contamination.
(1)$$P_{N}=\sqrt{\frac{({C_{i}/D_{i})}_{max}^{2}+{\left(C_{i}/D_{i}\right)}_{ave}^{2}}{2}}$$
where (C_i_/D_i_)_max_ is the maximum value of the pollution index of soil heavy metal elements; (C_i_/D_i_)_ave_ is the average value of the pollution index of soil heavy metal elements. Grading criteria: P_N_ ≤ 0.7, safe; 0.7 < P_N_ ≤ 1, warning line; 1 < P_N_ ≤ 2, mild pollution; 2 < P_N_ ≤ 3, moderate pollution; P_N_ > 3, severe pollution.

Plant samples were washed with distilled water to remove traces of surface elements, then divided into roots and shoots and oven-dried at 65 °C for 48 h to constant weight. Then, each sample was crushed to a fine powder with a mortar and pestle, and 0.2 g roots/shoots powders were digested with 5 mL HNO_3_ (65%) at 110 °C for 2 h, then cooled and mixed with 1 mL H_2_O_2_ (30%, *v*/*v*) and boiled for 1 h. Finally, the digests were diluted to 50 mL with triple deionized water in a volumetric flask. Triplicates were prepared for each sample. The concentrations of Pb and Cd in the samples were determined by flame atomic absorption spectrometry [41]. The quality control was conducted using the GBW100348 plant standard material of the National Institute of Metrology, China [42]. The Biological Concentration Factor (BCF) of each organ of the plant and Transfer Factor (TF) were calculated according to the following equations [43,44]:(2)$$BCF=\frac{C_{aerialpart}}{C_{soil}}$$
(3)$$TF=\frac{C_{aerialpart}}{C_{undergroundpart}}$$
where C_aerial part_ (mg/kg) represents the metal concentration of stems and leaves of the plant, C_soil_ (mg/kg) is the soil metal concentration, and C_underground part_ represents the metal concentration of roots and whips of the plant.

### 4.3. DNA Extraction, PCR Amplification, and Illumina Miseq Sequencing

The CTAB method was used to extract total DNA from plant tissues [45]. The fresh plant stems underwent surface sterilization by soaking in 7% NaClO for 3 min and 75% NaClO for 5 min. The residual chemicals were removed by washing the tissues with deionized water several times. Approximately 1 g of surface-sterilized *Artemisia argyi H. Lév. & Vaniot.* stems were frozen with liquid nitrogen and ground to a fine powder in a sterilized and precooled mortar. The *Artemisia argyi H. Lév. & Vaniot.* plant DNA was extracted using the EasyPure Plant Genomic DNA kit (Transgen Biotech, Beijing, China) according to the manufacturer’s protocols. The 16S rDNA target region of the ribosomal RNA gene was amplified via PCR (pre-denaturation at 95 °C for 3 min, 95 °C for 5 min, followed by 27 cycles at denaturation at 95 °C for 30 s, annealing at 55 °C for 30 s, and extension at 72 °C for 30 s) using 341F primer (5′-ACTCCTACGGGAGGCAGC-3′) and 806R primer (5′-GGACTACNVGGGGTWTCTAA-3′)) targeting the V3–V4 region. In total, 20 μL PCR mixture containing 4 μL of 5×Trans Start Fast Pfu Buffer (Trans Start, Beijing, China), 2 μL of 2.5 mmol/L deoxyribonucleoside triphosphates, d NTPs, 0.8 μL of upstream primer (5 μmol/L), 0.8 μL of downstream primer (5 μmol/L), 0.4 μL of Trans Start Fast Pfu DNA polymerase, 10 ng of template DNA, and double-distilled water (dd H_2_O) were combined up to 20 μL. The PCR amplification products of the three replicates were mixed, and the products were detected using 2% agarose gel electrophoresis. The second-round PCR amplified products were purified using the Axy Prep DNA Gel Extraction Kit (Axygen, Hangzhou, China) and then quantified by Quantus^TM^ Fluorometer (Promega, Beijing, China) detection and mixed according to the sequencing volume required for each sample in the appropriate proportion. High-throughput sequencing by the Illumina MiSeq PE300 platform (Revvity, Beijing, China) was performed.

The raw data obtained from sequencing were processed by trimming, filtering, and splicing to obtain valid data for subsequent analysis. The optimized sequences, showing >97% similarity, were clustered into operational taxonomic units (OTUs). Sequence data were processed, and bioinformatics was analyzed using QIIME (v1.9.1), as previously described by Hakim [46]. Analyses of the 16S rRNA sequencing data were performed on the Omicsmart Platform (www.omicsmart.com, accessed on 15 February 2024). Alpha-diversity indices (including ACE, Chao1, Shannon, and Simpson indices) were quantified based on OTU abundance. Demonstration of dissimilarity distances across samples was based on the Bray–Curtis distance using Principal Coordinates Analysis (PCoA) and non-metric multidimensional scaling (NMDS). The stacked bar plot of the community composition was visualized in the R project ggplot2 package. Analysis of function difference between groups was calculated by Tukey’s HSD test in the R project Vegan package (version 2.5.3). Association analysis and visualization of species with environmental factors used Pearson correlation network graphs. KEGG function prediction used Tax4Fun for SILVA annotation of 16S sequences based on prediction combined with heatmaps to visualize the differences in the functional distribution of samples. The endophytic bacterial sequences of plants from different sample areas have been submitted to the GenBank database (BioProject ID:PRJNA1097327).

### 4.4. Isolation, Selection, and Identification of Endophytic Bacteria

The fresh roots of *Artemisia argyi H. Lév. & Vaniot.* plants were rinsed with tap water, disinfected with 75% alcohol (30 s), and then with 10% NaClO (1 min), and rinsed five times in distilled water. The sterile roots (10 g fresh weight) were macerated using a sterile mortar and pestle. The macerated tissue (1 g) was transferred into plastic tubes with 9 mL sterile water and then serially diluted (10^−1^–10^−5^). In total, 100 µL aliquots from the dilutions (10^−3^, 10^−4^, 10^−5^) were spread on a Nutrient Agar (NA) medium, and plates were incubated for 72 h at 28 °C. Once the colonies were visible, an inoculation ring was used to select a single colony, which was then purified and cultured using the streaking method. This purification process was repeated three times to ensure that only one strain was grown in each medium. The resulting strain was stored in a refrigerator at 4 °C.

This paper aims to screen high-strength plant endophytes that are resistant to Pb and Cd. To determine heavy metal resistance, the NB (Nutrient Broth) medium was used as the basic medium. The culture was shaken at 150 r/min for 24 h at 30 °C, and 100 mL of heavy metal NB liquid medium with varying concentrations was prepared. The strain was cultured in a shaker at 28 °C and 200 r/min for 48 h after placing the 0.2 mL seed solution in a conical flask. Three parallel experiments were conducted to observe the strain’s growth. The JG1 strain with the best growth condition was selected as the research subject after screening for stress resistance. The bacteria were identified using morphological characteristics and 16S rRNA analysis. The obtained sequence was uploaded to the Genbank database (http://www.ncbi.nlm.nih.gov/, accessed on 20 March 2024 BioProject ID:OQ626703), and its similarity to the published sequence in the Genbank database was determined by BLAST.

### 4.5. Growth-Promoting Benefits of Endophytic Bacteria JG1

#### 4.5.1. Bacterial Agent Preparation

The JG1 strains were inoculated into NA medium and cultured for 3 days (28 °C). Afterward, the colonies were picked and added to 30 mL of LB (Luria Bertani) broth (10 g peptone, 5 g yeast, 5 g NaCl, 1000 mL deionized water, pH 7.2), incubated at 28 °C, and 200 r/min for 12 h. The bacterial suspension’s absorbance was measured at 600 nm (UV-8000 T, Shanghai Metash Instruments Co., Ltd., Shanghai, China). The OD_600_ values of the bacterial broth were adjusted to fall within the range of 0.8 to 1.2 through dilution or continued fermentation. The solution is then loaded into a 50 mL sealed tube and stored in a refrigerator at 4 °C for later use.

#### 4.5.2. Experimental Design

*Amorpha fruticosa* L. was selected as the experimental plant. The seeds were sterilized before the experiment (75% alcohol 30 s, 1% NaClO 1 min, and rinsed five times in distilled water). Impurity-removed soil was sterilized in an autoclave (121 °C, 20 min). Healthy and similarly grown seedlings germinated in seedling trays for 30 days (25 °C and relative humidity of 60%) were transplanted into pots that had been standing for 30 days. The experiment was divided into four groups: no heavy metal group (Pb = 0, Cd = 0, CK, JG1) and heavy metal group (Pb = 268.84 mg/kg, Cd = 63.06 mg/kg, HM + CK, HM + JG1). When the seedlings were transplanted into the pot, 50 mL of JG1 bacterial solution was injected around the roots of the seedlings, and 50 mL of liquid medium was added to the control group. Each treatment was replicated three times.

#### 4.5.3. Determination of Potted Plant Indicators

After two months of inoculation, the plants were harvested. For plants, vernier calipers and tape were used to measure the ground diameter and height of the seedlings. Root scanners were used to measure the root volume and leaf area (ten upper, middle, and lower leaves were selected from each pot to measure the leaf area). The plants were dried to measure the aboveground and underground biomass. For potting soil, the Mettler toledo pH meter was used to measure the soil’s pH value (water–soil ratio was 5:1). The alkaline hydrolysis diffusion method was used to determine hydrolyzed nitrogen (HN). The molybdenum-antimony anti-colorimetric method with NaHCO_3_ extraction was used to determine available phosphorus (AP), and flame photometry with NH_4_OAc extraction was used to determine available potassium (AK).

### 4.6. Statistical Analysis

Data were analyzed by analysis of variance (ANOVA) with SPSS 20.0 (IBM, USA) and means compared using Tukey’s test, with significant differences noted at *p* < 0.05. All graphics were drawn using Microsoft Excel 2019 software and OriginPro 2024.

## 5. Conclusions

In this study, the endophyte communities of *Artemisia argyi H. Lév. & Vaniot.* grown under different concentrations of heavy metals were analyzed by high-throughput sequencing. The results indicate that the Cyanobacteria and Proteobacteria phyla and *Methylobacterium* and *Sphingomonas* genera play an important role in improving the tolerance of *Artemisia argyi H. Lév. & Vaniot.* to Pb and Cd pollution. PCoA and correlation analyses revealed that Cd and Pb concentrations were the primary factors causing the changes in the structure of microbial communities. Tax4Fun analyses showed that the abundance of microorganisms involved in the metabolic functions of plant endophyte communities was affected by the concentration of heavy metals in the environment. Furthermore, plate culture experiments and pot experiments were conducted to analyze the tolerance of the JG1 (*Serratia marcescens*) strain to heavy metals and its ability to promote *Amorpha fruticosa* L. growth and improve soil properties. The results showed that plants inoculated with JG1 had significantly higher height, diameter, and biomass compared to the plants in the control group. This study provided a deeper understanding of the response mechanism of *Artemisia argyi H. Lév. & Vaniot.* to Cd and Pb stress. Additionally, the JG1 (*Serratia marcescens*) strain identified in this study can aid in the development of a microbial-assisted phytoremediation strategy for heavy metal removal from contaminated soils. The findings of this study can be used for practical applications of phytoremediation in heavy metal-polluted areas.

## Figures and Tables

**Figure 1 plants-13-02755-f001:**
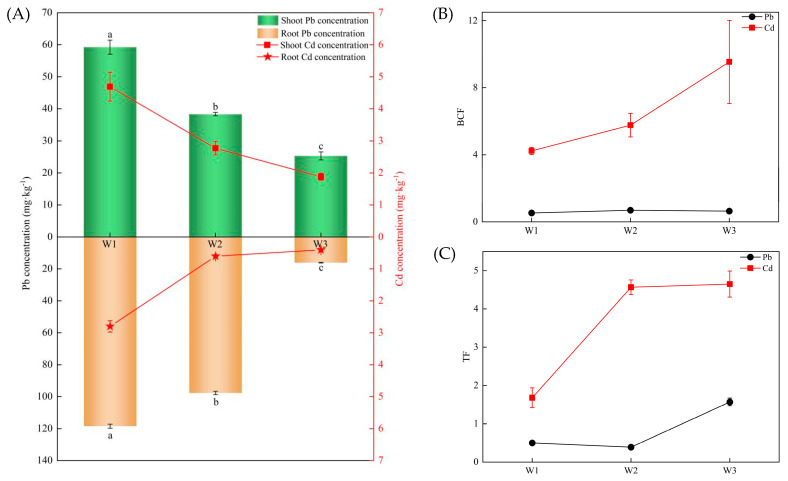
Mean contents of heavy metals in different parts of plants in W1, W2, and W3 sample plots (**A**), BCF (**B**), and TF (**C**).

**Figure 2 plants-13-02755-f002:**
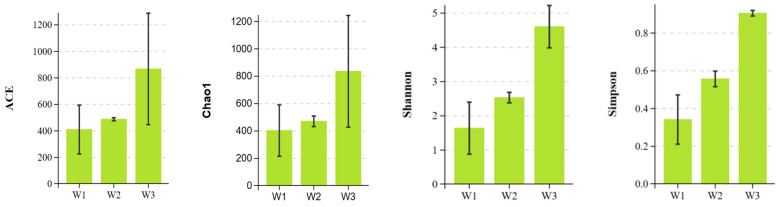
α-diversity of bacteria from the plants samples at distance < 0.03.

**Figure 3 plants-13-02755-f003:**
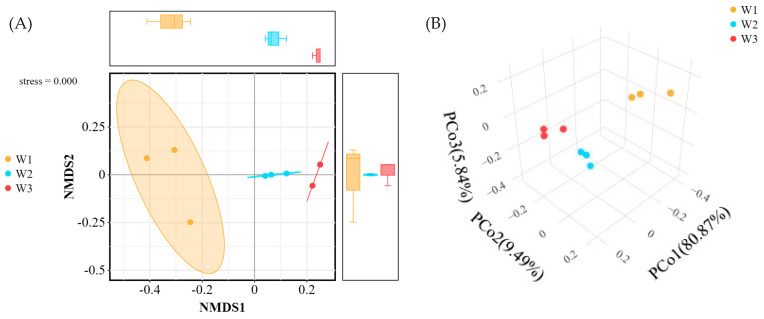
NMDS and PCoA based on the Bray–Curtis distance ((**A**) is the NMDS analysis of phytobacterial communities based on Bray-Curtis distance analysis between samples; (**B**) is the PCOA analysis).

**Figure 4 plants-13-02755-f004:**
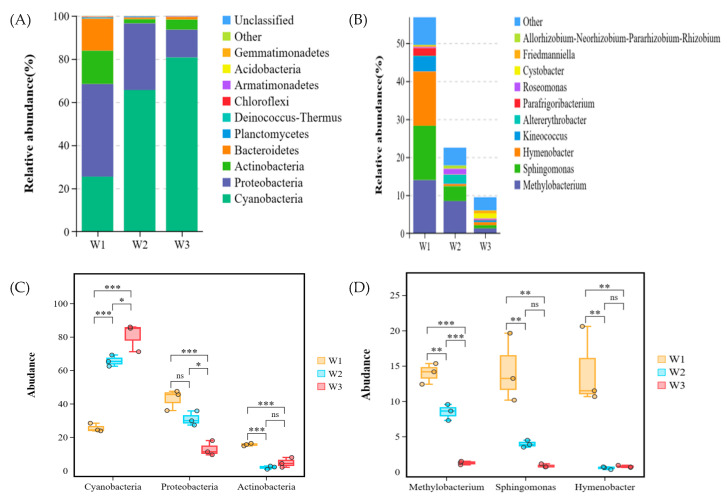
Taxonomic composition of dominant microbial communities in three habitats (W1, W2, W3) ((**A**): phylum level, (**B**): genus level) and Tukey’s HSD test at phylum (**C**) and genus (**D**) levels with 95% confidence intervals. *: *p* < 0.05, **: *p* < 0.01, ***: *p* < 0.001, ns: no significant difference.

**Figure 5 plants-13-02755-f005:**
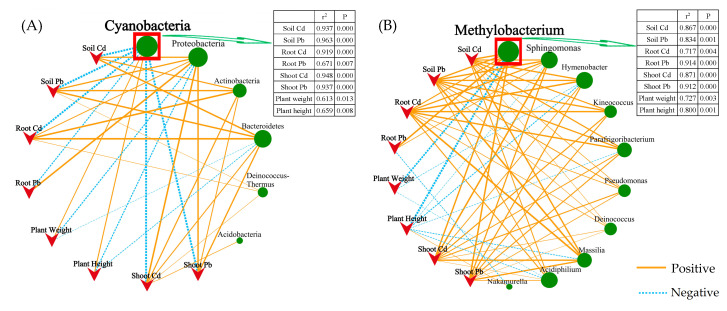
Correlation network map illustrating the correlation plant growth status and heavy metal content in plants with the phytobacterial community at the phylum (**A**) and genus (**B**) levels. Nodes represent genus and soil nutrient, lines represent correlations, and the thickness of the line represents the degree of correlation (*p* < 0.05).

**Figure 6 plants-13-02755-f006:**
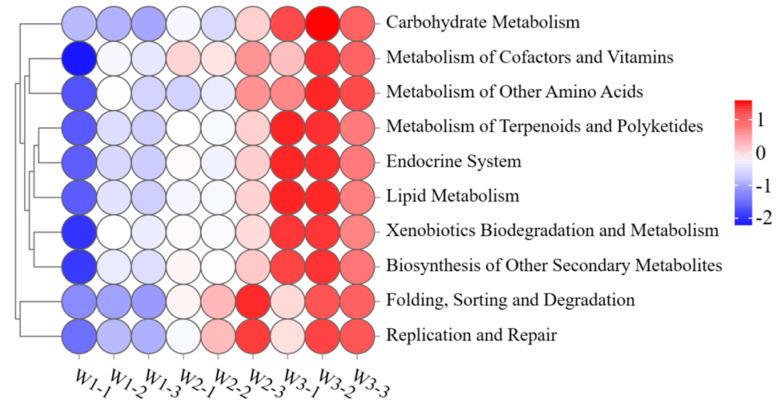
Heat map cluster and abundance of microbial functions at the genus level.

**Figure 7 plants-13-02755-f007:**
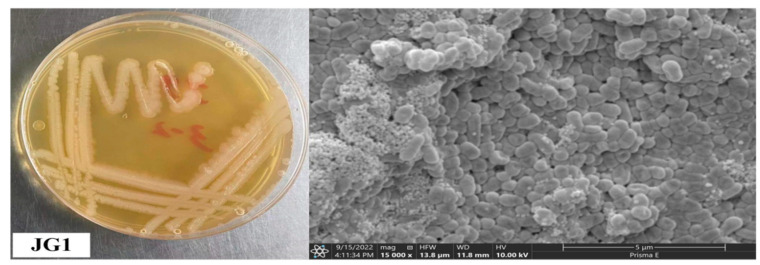
Colonies of JG (**left**) and cell morphology SEM images (**right**).

**Figure 8 plants-13-02755-f008:**
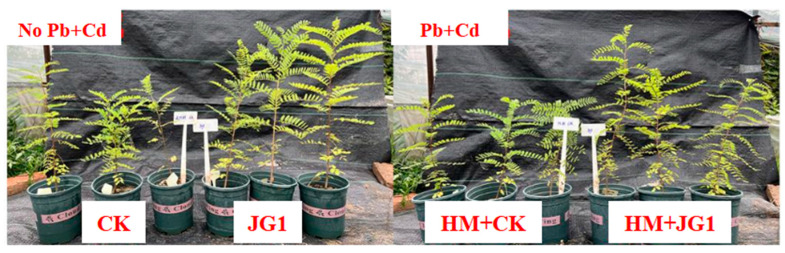
Effects of JG1 on plants.

**Figure 9 plants-13-02755-f009:**
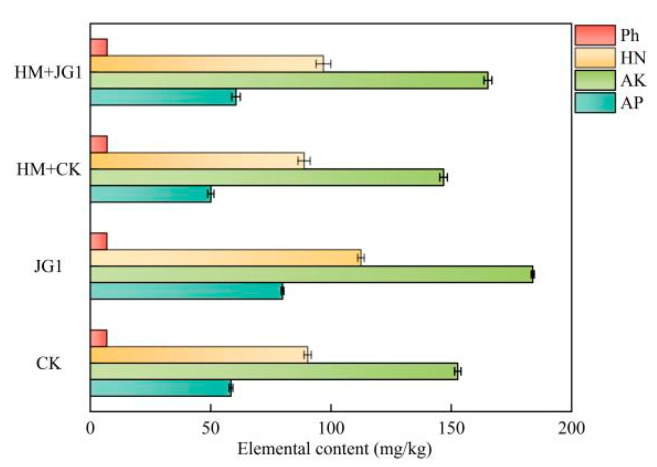
Effects of JG1 on soils (*p* < 0.05).

**Table 1 plants-13-02755-t001:** W1, W2, and W3 plot plant rhizosphere soil samples, Pb and Cd concentrations, and plant growth status.

Sites (Samples)	Soil Heavy Metal Content		P_N_	Plant Growth Indicator	
	Pb (mg/kg)	Cd (mg/kg)		Height (cm)	Weight (g)
W1	110.49 ± 2.34 a	1.11 ± 0.05 a	5.76	31.17 ± 3.83 b	2.39 ± 0.33 c
W2	55.06 ± 0.98 b	0.48 ± 0.06 b	2.81	37.36 ± 2.11 b	2.77 ± 0.32 b
W3	39.06 ± 2.12 c	0.20 ± 0.04 c	1.89	46.71 ± 4.65 a	3.54 ± 0.38 a
Background Value of Shanxi Province	14.70	0.102	2.3	-	-
Natural Background	170 ^1^	0.60 ^1^	-	-	-

Notes: Mean ± standard deviation from three samples. Different letters in the same column indicate a significant difference at *p* < 0.05. ^1^ means risk screening values of agricultural land in the soil environmental quality of China (GB 15618-2018).

**Table 2 plants-13-02755-t002:** Effects of JG1 on plants.

Groups	Plant (Aboveground)			Plant (Underground)		Plant Biomass (g)
	Plant Height (cm)	Ground Diameter (mm)	Average Leaf Area (cm^2^)	Root Length (cm)	Root Volume (cm^3^)	
CK	39.97 ± 3.96	7.38 ± 0.41 ab	1.60 ± 0.08 b	575.8 ± 181.61 c	3.8 ± 1.56 b	20.64 ± 2.63 b
JG1	61.63 ± 6.95 a	7.59 ± 0.75 a	2.75 ± 0.12 a	1208.6 ± 351.86 a	5.3 ± 1.49 a	27.63 ± 2.99 a
HM + CK	45.80 ± 5.27 b	5.58 ± 0.54 b	1.39 ± 0.26 b	867.86 ± 64.65 b	4.00 ± 1.01 b	22.67 ± 1.54 a
HM + JG1	56.40 ± 6.99 a	7.77 ± 0.48 a	2.22 ± 0.15 a	704.94 ± 141.61 b	2.45 ± 0.93 c	25.58 ± 1.28 a

Notes: Mean ± standard deviation from three samples. Different letters in the same column indicate a significant difference at *p* < 0.05.

## Data Availability

The authors confirm that the data supporting the findings of this study are available within the article.

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
