# Peer review of "Response of Plant Endophyte Communities to Heavy Metal Stress and Plant Growth Promotion by the Endophyte Serratia marcescens (Strain JG1)"

_plants, 2024, doi:10.3390/plants13192755_

Round 1
Reviewer 1 Report
Comments and Suggestions for Authors
The work submitted for review is written very clearly and by the requirements. The aim of the work has not been clearly defined, although it can be assumed that it is "To understand the response strategies of endophytic communities to varying concentrations of Pb and Cd pollution". It would be worth declaring the aim of the work more clearly. The methodology does not raise any objections, the presentation of results and their discussion with the literature are also correct. Minor deficiencies are listed below:
line # 56: Here you should specify what kind of endophytes - bacterial or fungal.
line # 64: Artemisia L. it is not a species description but only a genus. If you talk about species - you should specify their names correctly.
lines # 86 – 89: At what basis these concentrations were defined as severe, moderate, or mild? What kind of regulations, norms, official documents, etc.?
Author Response
评论1:在这里,您应该指定哪种内生菌 - 细菌或真菌。
回应 1:感谢您指出这一点。细菌内生菌。文本已被修改并标记为红色。
评论2:蒿 L. 它不是一个物种描述,而只是一个属。如果你谈论物种 - 你应该正确指定它们的名字。
回应 2:感谢您指出这一点。艾蒿 Argyi H. Lév.&瓦尼奥特。文本已被修改并标记为红色。
评论3:这些浓度被定义为重度、中度或轻度的依据是什么?什么样的法规、规范、官方文件等?
回应 3:感谢您指出这一点。 本研究采用《土壤环境质量农业土壤污染风险控制标准(试行)(GB15618-2018)》中农田土壤污染风险的筛选值和控制值作为农田重金属污染的评价标准。由于研究区pH测量结果大于7.5,本研究选取pH>7.5状态下农田重金属污染筛查控制值最低值作为场地重金属污染评价标准,同时结合山西省土壤元素背景值, 评判和进一步完善评估过程;最后,基于Nemerow综合污染指数的大小来反映研究区表层土壤的重金属污染程度。

Reviewer 2 Report
Comments and Suggestions for Authors
The MS entitled “Response of plant endophyte communities to heavy metal stress and plant growth promotion by the endophyte Serratia marcescens (strain JG1)” can be accepted but some serious points need to be corrected.
- L.10 “H” please add lowercase in “hovewer”
- L.84 “Study sites…” please add capital “S”
- Paragraph 2.1 According to the text, only one year (2021) is reported for in field sampling?
A single year of open field sampling can be misleading; many factors could affect the result. In the discussion/conclusions sections this aspect should be highlighted underlining that further replies should be conducted.
- L.95 “25°C” add space between the number and the symbol “°C”
Please, correct throughout the text also for the other units of measurement.
- L.196 remove the italic from the “L” of Linneo. Check the text.
- L.196 remove the double point between “plant. The seeds...”
- Figure 5. Please correct “negative” in the figure key.
- L. 316 “0.05<p<0.5” is this correct? P<0.5?
- On what basis only 12 of the 30 isolates were chosen? Please, explain your choice.
Author Response
Comments 1: “H” please add lowercase in “hovewer”; “Study sites…” please add capital “S”; “25°C” add space between the number and the symbol “°C”; remove the italic from the “L” of Linneo. Check the text; remove the double point between “plant. The seeds...”; “0.05<p<0.5” is this correct? P<0.5?
Response 1:Thank you for pointing this out, I have corrected the errors in the text and marked it in red.
Comments 2: Paragraph 2.1 According to the text, only one year (2021) is reported for in field sampling?
Response 2: Thank you for asking this question. This study focused on the degree of heavy metal pollution in different ranges of slope tailings, and screened strains with stress resistance and growth promotion characteristics from heavy metal contaminated soil, so the sampling time was only one year.
Comments 3: On what basis only 12 of the 30 isolates were chosen? Please, explain your choice.
Response 3: Thank you for asking this question. 12 strains with growth promoting effect were screened out through the plate culture experiments of dissolving phosphorus, fixing nitrogen and dissolving potassium.

Round 2
Reviewer 2 Report
Comments and Suggestions for Authors
Please, remove the italic from the “L” of Linneo. Check the text.
Author Response
Comments 1:Please, remove the italic from the “L” of Linneo. Check the text.
Response 2:Thank you for pointing this out. The text has been modified and marked in red.